# Association between Dietary Choline Intake and Cardiovascular Diseases: National Health and Nutrition Examination Survey 2011–2016

**DOI:** 10.3390/nu15184036

**Published:** 2023-09-18

**Authors:** Rong Zhou, Mei Yang, Chaofu Yue, Yi Shi, Yanan Tan, Lingfeng Zha, Junxia Zhang, Shaoliang Chen

**Affiliations:** 1Department of Intensive Medicine, Qujing First People’s Hospital, Qujing 655000, China; 220193772@aa.seu.edu.cn (R.Z.); yangmei@kmmu.edu.cn (M.Y.); yueaaa0@163.com (C.Y.); 2Department of Cardiology, Nanjing First Hospital, Nanjing Medical University, Nanjing 210006, China; shiyi@stu.njmu.edu.cn (Y.S.); nan_072300@163.com (Y.T.); 15850770739@126.com (S.C.); 3Department of Cardiology, Union Hospital, Tongji Medical College, Huazhong University of Science and Technology, Wuhan 430022, China; 4Hubei Key Laboratory of Biological Targeted Therapy, Union Hospital, Tongji Medical College, Huazhong University of Science and Technology, Wuhan 430022, China; 5Hubei Provincial Engineering Research Center of Immunological Diagnosis and Therapy for Cardiovascular Diseases, Union Hospital, Tongji Medical College, Huazhong University of Science and Technology, Wuhan 430022, China

**Keywords:** dietary choline intake, cardiovascular disease, NHANES

## Abstract

Choline is an essential nutrient for human body, but dietary choline is metabolized into the hazard metabolite for the cardiovascular system. Because of the conflicting results between dietary choline intake and cardiovascular disease (CVD) risk in previous studies, we aimed to investigate this in US adults. Non-pregnant participants and those aged >20 years from National Health and Nutrition Examination Survey 2011–2016, with CVD assessment and reliable dietary recall status, were included. The dietary choline intake was assessed as a mean value of two total dietary choline intakes, including dietary choline intake and supplemental choline intake, in 24-h dietary recall interviews. The association between dietary choline intake and the presence of CVD was examined using logistic regression. We enrolled 14,323 participants. The participants without CVD had substantially higher dietary choline intakes (318.4 mg/d vs. 297.2 mg/d) compared to those with CVD (*p* < 0.05). After multivariable adjustments, the highest quartile of dietary choline intake was associated with a lower CVD risk, OR 0.693, 95%CI [0.520, 0.923], when compared to the lowest quartile. Consistent results were also found for stroke. Subgroup analyses also supported these, especially in participants aged ≥60 years and in those with BMI < 30 kg/m^2^. We found that a higher dietary choline intake was associated with a lower CVD risk, especially the risk of stroke. Further clinical trials are needed in order to confirm this finding and to provide dietary suggestions for the appropriate amount of choline intake.

## 1. Introduction

Cardiovascular diseases (CVD), including chronic heart failure (CHF), coronary heart diseases (CHD), and stroke, are considered the primary cause of death worldwide [1]. Globally, 17.3 million people die from CVD each year, and that number is rising, placing a significant burden on the medical care system [2]. Therefore, it is of the utmost importance to reduce CVD incidence by controlling related risk factors as soon as possible. Choline is an essential ingredient that is required for many biological processes in the human body, including the formation of cell membranes, the preservation of liver and kidney function, and the production of neurotransmitters [3]. For humans, only a small amount of choline can be endogenously generated through the liver phosphatidylethanolamine N-methyltransferase pathway. It is vital to supplement it in the diet to prevent deficiency. Many foods, such as red meat, eggs, fish, green vegetables, and whole grains, are sources of choline. Nevertheless, the gut microbiome can convert dietary choline into trimethylamine, which is then metabolized to trimethylamine-n-oxide (TMAO) by the flavin monooxygenases enzymes in the liver [4]. A recent collection of evidence has suggested that TMAO could promote the development of adverse cardiovascular events [5]. A study found an increased risk of dietary consumption of choline causing all-cause and CVD mortality in the Nurses’ Health Study (1980–2012) and the Health Professionals Follow-Up Study (1986–2012) [6]. This result was later included in a meta-analysis involving 18,076 incident CVD events and 5343 CVD deaths from 184,010 total participants. However, the meta-analysis found there was no association between choline and incident CVD [7]. Considering the choline provided by food is regarded as a paramount necessity for humans, it is important to determine dietary advice related to choline. Herein, we investigated the association between dietary choline intake and CVD risk through additional data from the 2011–2016 cross-sectional National Health and Nutrition Examination Survey (NHANES).

## 2. Materials and Methods

### 2.1. Study Population

The NHANES was a survey conducted by the US Centers for Disease Control and Prevention (CDC) to monitor the health and nutrition statuses of the adults and children in the US. The survey was approved by the Institutional Review Board of the National Center for Health Statistics (NCHS), and all participants gave informed consent. NHANES used “stratified multistage probability sampling” to obtain a nationally representative sample. The details of survey administration and methods can be accessed on the NHANES website (http://www.cdc.gov/nchs/nhanes.htm, accessed on 18 March 2023).

We combined 6 consecutive NHANES circles from 2011/2012 to 2015/2016 according to the NHANE analytical guidelines. The exclusion criteria were as follows: (1) participants aged ≤20 years, (2) participants who were pregnant, (3) participants missing an assessment of CVD, (4) participants without reliable dietary recall status. The selection process was shown in Figure 1. Finally, 14,323 participants were included in the subsequent analysis.

### 2.2. Assessment of CVD

The CVD outcomes were diagnoses self-reported in a validated medical condition questionnaire during the personal interview, and participants were asked the following questions: “Has a doctor or other health professional ever told you that you have CHF/CHD/angina pectoris/heart attack/stroke?” Those who answered “yes” to the any of the questions were labeled as positive for CVD. In our study, we combined CHD, angina pectoris, and heart attack in the questionnaire regarding CHD, and the CVD outcomes included three types: CHF, CHD, and stroke.

### 2.3. Assessment of Dietary Choline Intake

Dietary choline intake was obtained from two 24-h dietary recall interviews, according to Food and Nutrient Database for Dietary Studies of United States Department of Agriculture. The two 24-h dietary recalls were collected by trained interviewers using the validated automated multiple pass Method. The first dietary recall interview was conducted in person at the Mobile Examination Center (MEC), and the second interview was conducted by telephone 3 to 10 days later. After the 24-h dietary recall, 24-h dietary supplement usage was collected, and the supplement amounts were obtained from NHANES’ dietary supplement database. In this study, the total choline intake included dietary intake and supplemental intake over 24 h, and we took the average of two days of total choline intake as the actual choline intake.

### 2.4. Covariates

Covariates in this study consisted of demographic characteristics (gender, age, race, education level, marital status, poverty/income ratio (PIR), physical activity category, body mass index (BMI), and smoking and drinking status), medical history (diabetes mellitus [DM], and hypertension), and laboratory results (white blood cells (WBCs), lymphocyte cells, monocyte cells, neutrophil cells, eosinophil cells, basophil cells, platelets (PLTs), red blood cells (RBC), hemoglobin (Hb), albumin (Alb), alanine transaminase (ALT), aspartate transaminase (AST), blood urea nitrogen (BUN), creatinine (Cr), uric acid (UA), estimated glomerular filtration rate (eGFR), glycohemoglobin (HbA1c), apolipoprotein B (Apo B), high-density lipoprotein cholesterol (HDL-C), triglyceride, and low-density lipoprotein cholesterol (LDL-C)).

The race categories included Mexican American, other Hispanic, non-Hispanic White, non-Hispanic Black, and other. Education levels were classified into below high school, high school, and above high school. Marital status was divided into three categories: married, separated, and never married. The PIR was calculated by dividing family (or individual) income by the poverty guidelines specific to the survey year, and a higher PIR indicated a better family income status. In this study, PIR was stratified as 0–1, 1–3, and >3 according to the original value. Weight and height were obtained from physical examinations, and BMI was calculated as weight in kilograms divided by height in meters squared. Participants who had smoked <100 cigarettes in their entire lives were defined as never smokers, those who had smoked ≥100 cigarettes but did not smoke at the time of survey were defined as former smokers, and those who had smoked ≥100 cigarettes and still smoked at the time of survey were defined as current smokers. Drinking status was categorized as non-drinker, low-to-moderate drinker (drinking <1 drink/day in women and <2 drinks/day in men), and heavy drinker (drinking ≥1 drink/day in women and ≥2 drinks/day in men) [8]. The terms of physical activity included work activity, walking or bicycling activity, and recreational activity. The weekly metabolic equivalent (MET) minutes of physical activity were recorded, and they were categorized as follows according to established standards: (1) below, i.e., <600 MET min/week or 150 min/week of moderate-intensity exercise; (2) meet, i.e., 600 to 1200 MET min/week or 150 to 300 min/week of moderate-intensity exercise; and (3) exceed, i.e., ≥1200 MET min/week or 300 min/week of moderate-intensity exercise [9]. DM was defined as a self-reported history of diabetes, HbA1c ≥ 6.5%, fasting plasma glucose ≥ 126 mg/dL, 2 h glucose in oral glucose tolerance test (OGTT) ≥200 mg/dL, taking antihyperglycemic agents, or taking insulin [10]. Hypertension was defined as a self-reported history of hypertension, blood pressure ≥ 140/90 mmHg, or taking antihypertensive agents [11]. The eGFR was calculated according to the four-variable Modification of Diet in Renal Disease (MDRD) [12]. The methodology for the laboratory tests is described in detail on the NHANES website (http://www.cdc.gov/nchs/nhanes/index.htm, accessed on 18 March 2023).

### 2.5. Statistical Analysis

The statistical analysis was performed in accordance with the analytic guidelines proposed by CDC. Considering the complex, multistage, probability sampling design of the NHANES, we integrated sample weights, stratification, and clustering into the statistical analysis. Continuous variables were presented in the form of weighted mean and standard error (SE), then compared using a survey-design-based *t*-test. Categorical variables were expressed by weighted percentages and compared using Rao-Scott χ^2^ statistics. The Proc Survey logistic regression analyses were applied to estimate the odds ratios (ORs) and 95% confidence intervals (CIs) for the dietary choline intake and incidence of CVD. The dietary choline intake was categorized by quartiles, and the lowest quartile was used as the reference. In crude model 1, no covariates were included, and in model 2, age, gender, race, and BMI were included as covariates. In model 3, age, gender, race, BMI, education level, marital status, smoking status, drinking status, physical activity category, hypertension, DM, and PIR were included as covariates. Restricted cubic spline (RCS) regression was conducted with 4 default knots located at the 5th, 35th, 65th, and 95th percentiles of dietary choline intake to test the linearity/nonlinearity of the association between choline and CVD. We also explored the associations between dietary choline intake and three subtypes of CVD in model 3. Subgroup analyses, stratified by age, gender, and BMI, were performed among the different populations with the same adjustments as in model 3. All statistical analyses were conducted using R software version 3.6.0 (R Core Team, 2022, Vienna, Austria; version 4.1.6). *p*-value < 0.05 was considered as statistically significant.

## 3. Results

### 3.1. Baseline Characteristics of the Study Participants

A total of 14,323 participants from NHANES (2011–2016) were included in this study (Figure 1). The characteristics of the participants are presented in Table 1. The mean dietary choline intake was 316.5 ± 164.1 mg/d, and the incidence of CVD was 8.8% in the study participants. In Table 1, according to the quartiles of dietary choline intake, participants were equally classified into four groups. Participants in the Q4 group were more likely to be male and low-to-moderate or heavy drinkers, and they engaged in more physical activity. In addition, high dietary choline intake was associated with a slower heart rate; elevated levels of monocyte count, RBC, Hb, Alb, ALT, AST, BUN and UA; as well as lower levels of WBC and PLT. Table 2 summarizes the baseline characteristics of the participants with and without CVD.

### 3.2. Association between Dietary Choline Intake and CVD

The association between the dietary choline intake and the presence of CVD are shown in Table 3. When compared with lowest quartile of dietary choline intake, the highest quartile was also associated with a lower incidence of CVD risk in all models: OR 0.728, 95%CI [0.580, 0.914] in unadjusted Model 1; OR 0.611, 95%CI [0.467, 0.799] in Model 2, adjusted for age, gender, race, and BMI; and OR 0.693, 95%CI [0.520, 0.923] in Model 3, adjusted for age, gender, race, BMI, education level, marital status, smoking status, drinking status, physical activity category, hypertension, DM, and PIR, respectively. The multivariable-adjusted RCS plot revealed that there was a linear association between dietary choline intake and the incidence of CVD (*p*-overall < 0.0001) (Figure 2). Table 4 shows the associations between dietary choline intake and subtypes of CVD. For different subtypes of CVD, a similar trend was found for stroke.

### 3.3. Subgroup Analysis of Dietary Choline Intake and CVD

A subgroup analysis, stratified by age, gender, and BMI, was carried out in this study (Table 5). For participants aged ≥60 years, high dietary choline intake in the Q4 group was negatively associated with the incidence of CVD: OR 0.669, 95%CI [0.479, 0.934], *p* = 0.020. Interestingly, for participants with BMIs < 30 kg/m^2^, the protective effect of high dietary choline against CVD was preserved, whereas for participants with BMIs ≥ 30 kg/m^2^, the protective effect was underpowered.

## 4. Discussion

The new findings of our study are as follows. (1) In contrast to previous studies, higher dietary choline intake was associated with a lower incidence of CVD, especially the incidence of stroke, in this large, nationally representative US population. (2) The protective role of higher dietary choline intake was accompanied by reduced inflammation and heart rate. (3) In the subgroup study, higher dietary choline intake in participants aged ≥60 years, and in participants with BMIs < 30 kg/m^2^, was found to be a protective factor for the presence of CVD. In summary, our results suggest that adequate choline intake acts against CVD and choline deficiency should be avoided.

There have been inconsistent findings regarding the relationship between dietary choline intake and CVD incidence, according to previous studies. Dalmeijer et al. [13] and Zheng et al. [6] found there was no association between dietary choline intake and incident CVD in post-menopausal Dutch women and in US women and men, respectively. The data from the Atherosclerosis Risk in Communities Study (ARIC) confirmed this [14]. A meta-analysis of four studies further supported that dietary choline was not associated with incident CVD (relative risk [RR] 1.00, 95% CI [0.98, 1.02]) [7]. However, data from 3924 Jackson Heart Study (JHS) showed a significant inverse association between dietary choline intake and incident stroke, β = −0.33 (*p* = 0.04) [15]. A cohort study involving 2606 subjects who were followed up for 10.6 years indicated that there was no association between total choline and the risk of CVD among adults, but a higher intake of free choline was associated with a lower risk of CVD (hazard rate [HR] 0.64, 95% CI [0.42, 0.98]) [16]. A PREDIMED-Plus trial found a longitudinal relationship between increased intake of dietary choline and improvements in cardiometabolic parameters [17]. Unlike the JHS study, which merely included African-American participants, and the PREDIMED-Plus trial, which evaluated cardiometabolic parameters, our study found for the first time that there was a protective effect of higher total choline intake on CVD incidence in 14,323 multi-ethnic participants.

Dietary factors play an important role in reducing cardiovascular disease risk, and some food sources rich in antioxidants and with anti-inflammatory, hypolipidemic, and hypoglycemic properties are thought to protect against the development of CVD. There is a complex association between dietary choline intake and CVD, which might be related to food source, TMAO concentration, and other factors. The PREDIMED-Plus trial suggested that food sources of choline might show a shift towards a healthier diet, including more fiber, omega-3 fatty acids, and polyphenols [17]. These foods might act synergistically with choline to improve the blood lipid profile and cardiometabolic parameters. Studies have indicated that TMAO concentration is primarily determined by the dietary consumption of precursors, gut microbial flora, liver flavin monooxygenase activity, and kidney function [18]. Hence, after taking into consideration the differences in TMAO generation, we determined that the association between higher dietary choline intake and CVD risk might be negative. In addition, according to the National Academies of Medicine, the recommended adequate intake (AI) of choline is 550 mg/day in adult men and 425 mg/day in adult women [19]. In our study, the mean dietary choline intake for participants was 316.5 mg/d, which was below the corresponding dietary choline recommendation. Choline is involved in the formation of specific phosphatidylcholine species, such as endogenous peroxisome proliferator-activated receptor α ligand (PPAR-α). PPAR-α participates in fatty acid oxidation, gluconeogenesis, lipid transport, and ketogenesis [15]. With inadequate dietary choline for phosphatidylcholine synthesis, the production of very low-density lipoproteins would decrease in the liver, whereas triglycerides would accumulate in the liver, heart, and arterial tissues, increasing the risk of CVD [20]. Hence, a sufficient dietary choline intake might be protective against the risk of CVD.

Several studies have demonstrated that dietary choline is independently associated with a reduction in inflammation mediators such as CRP, interleukin-6 (IL-6), and tumor necrosis factor-α (TNF-α) [21,22]. These inflammation mediators play crucial roles in CVD. Studies have shown that dietary choline intake could attenuate inflammatory responses by increasing S-adenosylmethionine (SAM) and reducing S-adenosylhomocysteine (SAH) [22,23]. The elevated SAM could inhibit inducible nitric oxide synthase and nuclear factor-B, and increase the production of glutathione [22]. The SAH could be converted to homocysteine, and the decreased homocysteine could contribute to the lower CVD risk [23]. In our analysis, we observed that dietary choline intake was inversely associated with WBC counts. Thus, dietary choline intake might protect against CVD by appeasing inflammation.

In our study, there was no obvious association of dietary choline intake with systolic or diastolic blood pressure, triglyceride, total cholesterol, or LDL-C. It was worth noting that a high dietary choline intake was significantly associated with a slowed heart rate in this study. The mechanism by which choline reduces the heart rate may be involved with enhanced endogenous production of a phosphatidylcholine molecule that is enriched in DHA. DHA is an ω-3 (*n*-3) long-chain polyunsaturated fatty acid that has been shown to reduce heart rate and improve vascular reactivity [24]. Moreover, as a substrate for acetylcholine, choline may relax vascular smooth muscles and reduce heart rate [25]. In general, HDL-C can neutralize the proinflammatory and pro-oxidant effects of monocytes to prevent atherosclerosis. The mechanism involves the inhibition of the migration of macrophages, oxidation of LDL, and efflux of cholesterol [26]. However, we found that higher dietary choline intake was associated with a lower HDL-C. This may arise from dietary choline intake collected at baseline and can be affected by multiple confounding factors, such as dietary covariates, total energy intake, and other lifestyle factors.

In this study, it was interesting to note that the protective effect of high dietary choline intake against CVD was significant in participants with BMIs < 30 kg/m^2^. Obese individuals were at a greater risk of numerous diseases, including DM, hypertension, and CVD. Considering lifestyle and dietary factors in the obese, the obese tended to engage in less exercise and to consume foods abundant in saturated fats and cholesterol. When dietary choline is mainly obtained through excessively high-fat foods, these unhealthy dietary habits might promote CVD. However, Donya’s study indicated that overweight/obese adolescents with higher dietary intakes of choline were less likely to be metabolically unhealthy [27]. Furthermore, we found that a higher dietary choline intake in participants aged ≥ 60 years was associated with lower incidence of CVD. Further studies are needed in order to confirm these subgroup findings.

There are some strengths of this study. Firstly, our study sample included a large and nationally representative sample of US adults. Secondly, the total choline intake was calculated by the average of total choline intake over two days, including dietary and supplemental intake per 24 h. Thirdly, considering multiple confounding factors, we performed the subgroup analyses stratified by age, gender, and BMI. Lastly, we found that reducing dietary choline intake to prevent CVD was not recommended. Our result provides new evidence for public dietary health.

There are several limitations of this study. Firstly, although we have adjusted many potential confounding factors, other unknown factors could not be completely ruled out, such as dietary covariates, total energy intake, and genetic factors. Secondly, dietary choline intake information was collected during two 24-h dietary recall interviews, but individual diet intake may change over time. Third, because the NHANES study was surveyed in the US population using the standardized health questionnaire, unavoidable recall and report bias may also have influenced the process of data acquisition. These are the limitations of the NHANES study.

## 5. Conclusions

We retrospectively analyzed 14,323 adults in the US from NHANES and found that higher dietary choline intake was associated with a lower CVD risk. Our current results may provide a new perspective on the correlation between dietary choline intake and CVD, but further studies are still needed in order to confirm our findings and titrate the appropriate amount of choline intake. In addition, pharmaceutical treatment is closely related to clinical outcomes of CVD, and future investigations could also consider to include pharmacological treatments.

## Figures and Tables

**Figure 1 nutrients-15-04036-f001:**
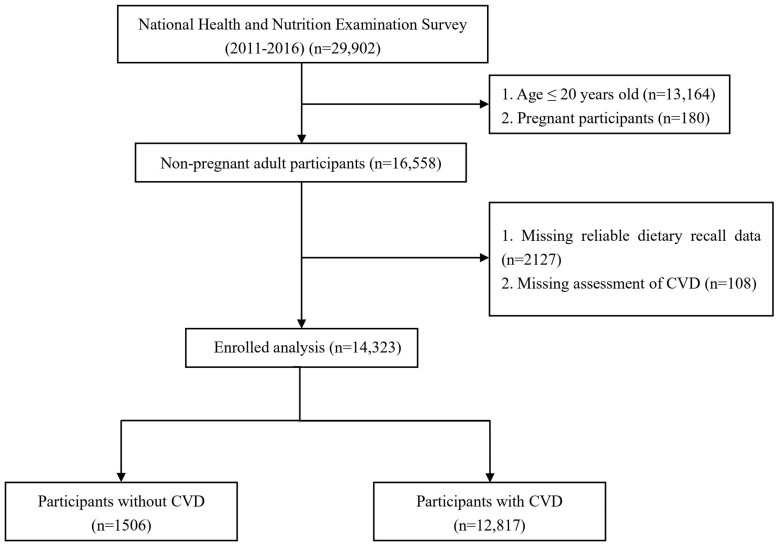
Flowchart of participants included in the analysis.

**Figure 2 nutrients-15-04036-f002:**
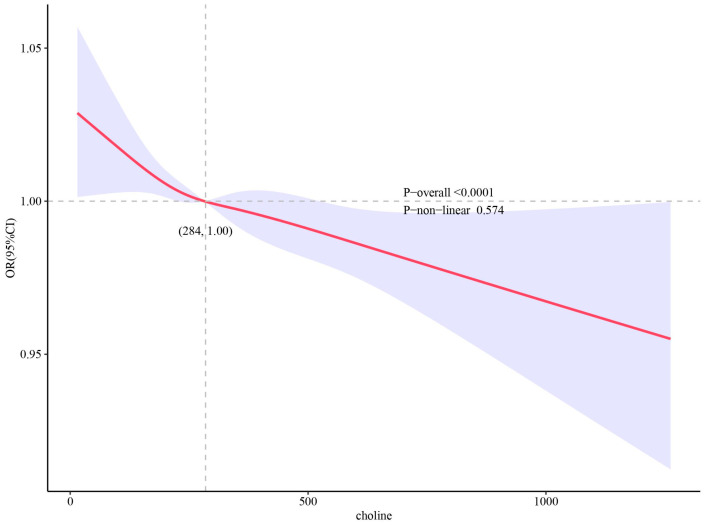
Restricted cubic spline (RCS) plot of the association between dietary choline intake levels and the presence of CVD. RCS regression was adjusted for age, gender, race, BMI, education level, marital status, smoking status, drinking status, physical activity category, hypertension, DM, and PIR (Model 3). The solid lines and shadow bands represent the OR and 95%CI.

**Table 1 nutrients-15-04036-t001:** Baseline characteristics of dietary choline intake among the general adult population in the National Health and Nutrition Examination Survey (NHANES), 2011–2016 (weighted).

Variables	Total	Quartiles of Dietary Choline Intake				*p*-Value
		Q1	Q2	Q3	Q4	
Participants, n	14,323	3566	3578	3583	3596	
Male (%)	7016 (48.8)	1226 (32.2)	1455 (40.1)	1806 (49.9)	2529 (70.8)	<0.001
Age (mean ± SD) (years)	48 ± 17	47 ± 17	49 ± 17	50 ± 16	48 ± 16	<0.001
Race (%)						<0.001
Mexican-American	1936 (8.6)	455 (8.6)	445 (7.8)	479 (8.2)	557 (9.7)	
Other Hispanic	1529 (6.0)	419 (6.9)	410 (6.3)	358 (5.4)	342 (5.4)	
Non-Hispanic White	5598 (65.9)	1300 (61.9)	1427 (66.9)	1456 (68.1)	1415 (66.2)	
Non-Hispanic Black	3208 (11.1)	871 (13.0)	807 (11.3)	775 (10.2)	755 (10.2)	
Other Race	2052 (8.4)	521 (9.7)	489 (7.7)	515 (8.1)	527 (8.5)	
Education level (%)						<0.001
Below high school	3175 (15.0)	1006 (19.3)	811 (15.3)	686 (12.4)	672 (13.4)	
High school	3117 (20.9)	825 (23.4)	735 (20.6)	752 (19.7)	805 (20.1)	
Above high school	8031 (64.1)	1735 (57.2)	2032 (64.1)	2145 (67.9)	2119 (66.5)	
Marital status (%)						<0.001
Married	7327 (54.4)	1608 (46.5)	1797 (54.0)	1991 (59.8)	1931 (56.4)	
Separated	3141 (19.0)	938 (22.6)	837 (19.8)	738 (18.3)	628 (15.7)	
Never married	3855 (26.6)	1020 (30.9)	944 (26.2)	854 (21.9)	1037 (27.8)	
BMI (mean ± SD) (kg/m^2^)	29.14 ± 6.86	29.22 ± 6.86	29.06 ± 7.19	29.14 ± 6.74	29.16 ± 6.67	0.9
WBC (mean ± SD) (1000 cells/μL)	7.29 ± 2.24	7.46 ± 2.34	7.26 ± 2.25	7.20 ± 2.22	7.26 ± 2.17	0.01
Lymphocyte cells (mean ± SD) (1000 cells/μL)	2.14 ± 0.99	2.21 ± 0.94	2.14 ± 0.95	2.10 ± 1.15	2.11 ± 0.90	0.001
Monocyte cells (mean ± SD) (1000 cells/μL)	0.57 ± 0.21	0.56 ± 0.21	0.56 ± 0.19	0.57 ± 0.21	0.59 ± 0.22	0.001
Neutrophils cells (mean ± SD) (1000 cells/μL)	4.33 ± 1.68	4.44 ± 1.78	4.31 ± 1.68	4.28 ± 1.58	4.31 ± 1.68	0.077
Eosinophils cells (mean ± SD) (1000 cells/μL)	0.20 ± 0.16	0.20 ± 0.16	0.20 ± 0.16	0.20 ± 0.15	0.21 ± 0.16	0.282
Basophils cells (mean ± SD) (1000 cells/μL)	0.05 ± 0.06	0.05 ± 0.06	0.05 ± 0.06	0.05 ± 0.05	0.05 ± 0.06	0.478
PLT (mean ± SD) (1000 cells/μL)	237.6 ± 60.2	245.9 ± 62.8	239.3 ± 59.5	236.1 ± 60.8	230.0 ± 57.0	<0.001
RBC (mean ± SD) (million cells/μL)	4.67 ± 0.48	4.59 ± 0.48	4.64 ± 0.49	4.68 ± 0.47	4.76 ± 0.47	<0.001
Hb (mean ± SD) (g/dL)	14.15 ± 1.46	13.81 ± 1.51	14.04 ± 1.42	14.20 ± 1.42	14.51 ± 1.42	<0.001
Alb (mean ± SD) (g/dL)	4.33 ± 0.34	4.29 ± 0.35	4.33 ± 0.34	4.34 ± 0.34	4.37 ± 0.33	<0.001
ALT (mean ± SD) (U/L)	25.5 ± 22.3	23.8 ± 15.9	24.9 ± 32.0	25.7 ± 18.0	27.5 ± 19.4	<0.001
AST (mean ± SD) (U/L)	25.8 ± 16.1	25.2 ± 16.6	25.5 ± 17.7	25.7 ± 13.3	26.7 ± 16.7	0.013
BUN (mean ± SD) (mg/dL)	13.8 ± 5.5	12.8 ± 6.1	13.6 ± 5.2	14.0 ± 5.3	14.7 ± 5.3	<0.001
Cr (mean ± SD) (mg/dL)	0.89 ± 0.37	0.88 ± 0.55	0.87 ± 0.29	0.88 ± 0.27	0.92 ± 0.31	<0.001
UA (mean ± SD) (mg/dL)	5.44 ± 1.40	5.27 ± 1.41	5.35 ± 1.36	5.42 ± 1.42	5.68 ± 1.38	<0.001
HbA1c (mean ± SD) (%)	5.65 ± 0.96	5.67 ± 1.03	5.64 ± 0.92	5.65 ± 0.93	5.66 ± 0.96	0.656
Smoking status (%)						<0.001
Never smoker	8086 (55.5)	2075 (57.5)	2157 (58.5)	2012 (54.6)	1842 (51.7)	
Former smoker	3391 (25.1)	687 (19.4)	779 (23.1)	949 (29.1)	976 (28.1)	
Current smoker	2846 (19.4)	804 (23.1)	642 (18.4)	622 (16.3)	778 (20.2)	
Drinking status (%)						<0.001
Non-drinker	4563 (24.8)	1470 (33.0)	1275 (28.3)	1087 (23.8)	731 (15.3)	
Low-to-moderate drinker	8479 (63.3)	1871 (56.9)	2036 (61.2)	2173 (64.0)	2399 (70.3)	
Heavy drinker	1281 (11.8)	225 (10.1)	267 (10.5)	323 (12.1)	466 (14.4)	
eGFR (mean ± SD) (mL/min/1.73 m^2^)	99.4 ± 29.7	101.1 ± 30.2	98.5 ± 27.8	99.6 ± 33.6	98.7 ± 26.5	0.132
Physical activity category (%)						<0.001
Below	5707 (35.9)	1623 (40.7)	1500 (37.9)	1391 (35.7)	1193 (29.9)	
Meet	1536 (10.4)	358 (9.5)	410 (11.2)	428 (12.5)	340 (8.4)	
Exceed	7080 (53.7)	1585 (49.7)	1668 (51.0)	1764 (51.7)	2063 (61.7)	
Hypertension (%)	5340 (33.6)	1380 (33.7)	1352 (34.5)	1367 (34.5)	1241 (31.8)	0.355
DM (%)	9324 (62.8)	2478 (68.8)	2169 (56.4)	2329 (63.5)	2348 (63.1)	<0.001
PIR (%)						<0.001
0–1	3291 (16.0)	1030 (22.2)	832 (16.1)	717 (12.9)	712 (13.4)	
1–3	5857 (36.2)	1517 (39.7)	1448 (36.6)	1416 (34.4)	1476 (34.4)	
>3	5175 (47.9)	1019 (38.1)	1298 (47.3)	1450 (52.7)	1408 (52.2)	
HR (mean ± SD) (bpm)	73 ± 12	74 ± 12	73 ± 12	72 ± 12	72 ± 12	<0.001
SBP (mean ± SD) (mmHg)	123 ± 17	123 ± 18	122 ± 17	122 ± 16	123 ± 16	0.428
DBP (mean ± SD) (mmHg)	71 ± 12	71 ± 12	70 ± 12	71 ± 11	71 ± 12	0.333
Apo B (mean ± SD) (mg/dL)	91.2 ± 24.7	91.7 ± 24.8	89.6 ± 24.2	91.7 ± 24.2	91.8 ± 25.4	0.01
HDL-C (mean ± SD) (mg/dL)	53.7 ± 16.8	53.9 ± 18.0	54.8 ± 17.1	54.1 ± 16.4	52.0 ± 15.8	<0.001
Triglyceride (mean ± SD) (mg/dL)	187.5 ± 171.5	191.1 ± 172.9	179.3 ± 174.6	190.0 ± 166.1	189.9 ± 172.2	0.115
LDL-C (mean ± SD) (mg/dL)	102.6 ± 37.9	102.0 ± 38.0	101.8 ± 36.7	103.4 ± 38.3	103.2 ± 38.4	0.3
Total cholesterol (mean ± SD) (mg/dL)	193.7 ± 41.5	193.9 ± 41.6	192.3 ± 41.5	195.4 ± 41.4	193.1 ± 41.5	0.118

1. Values are weighted means ± SE for continuous variables or weighted percentages for categorical variables. 2. Choline: Q1: <197 mg/d, Q2: 197–282 mg/d, Q3: 282–392 mg/d, Q4: ≥392 mg/d. 3. PIR: poverty/income ratio; DM: diabetes mellitus; HR: heart rate; SBP: systolic blood pressure; DBP: diastolic blood pressure; BMI: body mass index; WBC: white blood cells; PLT: platelets; RBC: red blood cells; Hb: hemoglobin; Alb: albumin; ALT: alanine transaminase; AST: aspartate transaminase; BUN: blood urea nitrogen; Cr: creatinine; UA: uric acid; eGFR: estimated glomerular filtration rate; HbA1c:glycohemoglobin; Apo B: apolipoprotein B; HDL-C: high-density lipoprotein cholesterol; LDL-C: low-density lipoprotein cholesterol. 4. The significance of differences between quartiles is indicated by the *p*-value. A *p*-value < 0.05 was considered as statistically significant.

**Table 2 nutrients-15-04036-t002:** Characteristics of participants with/without CVD (weighted).

	Non-CVD	CVD	*p*-Value
n	12,817	1506	
Male (%)	6190 (48.4)	826 (53.2)	0.026
Age (mean ± SD) (years)	47 ± 16	65 ± 14	<0.001
Race (%)			<0.001
Mexican-American	1806 (8.9)	130 (5.1)	
Other Hispanic	1384 (6.1)	145 (4.0)	
Non-Hispanic White	4853 (65.4)	745 (71.1)	
Non-Hispanic Black	2844 (11.0)	364 (12.8)	
Other Race	1930 (8.6)	122 (7.0)	
Education level (%)			<0.001
Below high school	2721 (14.3)	454 (21.6)	
High school	2733 (20.4)	384 (26.3)	
Above high school	7363 (65.3)	668 (52.2)	
Marital status (%)			<0.001
Married	6589 (54.5)	738 (53.6)	
Separated	2573 (17.7)	568 (33.0)	
Never married	3655 (27.8)	200 (13.4)	
Choline intake (mean ± SD) (mg/d)	318.4 ± 165.5	297.2 ± 148.1	0.001
BMI (mean ± SD) (kg/m^2^)	28.99 ± 6.78	30.77 ± 7.47	<0.001
WBC (mean ± SD) (1000 cells/μL)	7.26 ± 2.16	7.55 ± 2.95	0.001
Lymphocyte cells (mean ± SD) (1000 cells/μL)	2.15 ± 0.79	2.05 ± 2.17	0.003
Monocyte cells (mean ± SD) (1000 cells/μL)	0.56 ± 0.20	0.63 ± 0.23	<0.001
Neutrophils cells (mean ± SD) (1000 cells/μL)	4.31 ± 1.68	4.59 ± 1.68	<0.001
Eosinophils cells (mean ± SD) (1000 cells/μL)	0.20 ± 0.15	0.24 ± 0.22	<0.001
Basophils cells (mean ± SD) (1000 cells/μL)	0.05 ± 0.06	0.05 ± 0.06	0.073
PLT (mean ± SD) (1000 cells/μL)	239.2 ± 59.3	220.3 ± 67.1	<0.001
RBC (mean ± SD) (million cells/μL)	4.68 ± 0.47	4.55 ± 0.52	<0.001
Hb (mean ± SD) (g/dL)	14.18 ± 1.45	13.87 ± 1.59	<0.001
Alb (mean ± SD) (g/dL)	4.35 ± 0.34	4.18 ± 0.35	<0.001
ALT (mean ± SD) (U/L)	25.48 ± 18.00	26.06 ± 48.10	0.686
AST (mean ± SD) (U/L)	25.66 ± 14.53	27.06 ± 27.66	0.114
BUN (mean ± SD) (mg/dL)	13.47 ± 5.00	17.32 ± 8.53	<0.001
Cr (mean ± SD) (mg/dL)	0.87 ± 0.34	1.07 ± 0.56	<0.001
UA (mean ± SD) (mg/dL)	5.40 ± 1.38	5.81 ± 1.59	<0.001
HbA1c (mean ± SD) (%)	5.60 ± 0.90	6.19 ± 1.31	<0.001
Smoking status (%)			<0.001
Never smoker	7477 (57.0)	609 (39.4)	
Former smoker	2845 (24.0)	546 (36.7)	
Current smoker	2495 (19.0)	351 (23.9)	
Drinking status (%)			<0.001
Non-drinker	3972 (24.1)	591 (32.6)	
Low-to-moderate drinker	7682 (63.9)	797 (57.6)	
Heavy drinker	1163 (12.0)	118 (9.8)	
eGFR (mean ± SD) (ml/min/1.73 m^2^)	101.2 ± 29.1	80.9 ± 29.2	<0.001
Physical activity category (%)			<0.001
Below	4839 (34.2)	868 (53.4)	
Meet	1381 (10.4)	155 (10.9)	
Exceed	6597 (55.4)	483 (35.7)	
Hypertension (%)	4203 (29.9)	1137 (72.2)	<0.001
DM (%)	8076 (60.9)	1248 (82.7)	<0.001
PIR (%)			<0.001
0–1	2872 (15.6)	419 (20.2)	
1–3	5143 (35.2)	714 (46.6)	
>3	4802 (49.3)	373 (33.2)	
HR (mean ± SD) (bpm)	73 ± 12	70 ± 12	<0.001
SBP (mean ± SD) (mmHg)	122 ± 17	129 ± 20	<0.001
DBP (mean ± SD) (mmHg)	71 ± 11	67 ± 14	<0.001
Apo B (mean ± SD) (mg/dL)	91.5 ± 24.7	88.4 ± 24.43	0.001
HDL-C (mean ± SD) (mg/dL)	54.1 ± 16.8	50.0 ± 16.1	<0.001
Triglyceride (mean ± SD) (mg/dL)	188.2 ± 173.0	181.0 ± 154.0	0.212
LDL-C (mean ± SD) (mg/dL)	103.5 ± 37.9	93.3 ± 36.5	<0.001
Total cholesterol (mean ± SD) (mg/dL)	195.1 ± 41.2	179.3 ± 42.1	<0.001

Values are weighted means ± SE for continuous variables or weighted percentages for categorical variables. CVD: cardiovascular diseases; PIR: poverty/income ratio; DM: diabetes mellitus; HR: heart rate; SBP: systolic blood pressure; DBP: diastolic blood pressure; BMI: body mass index; WBC: white blood cells; PLT: platelets; RBC: red blood cells; Hb: hemoglobin; Alb: albumin; ALT: alanine transaminase; AST: aspartate transaminase; BUN: blood urea nitrogen; Cr: creatinine; UA: uric acid; eGFR: estimated glomerular filtration rate; HbA1c:glycohemoglobin; Apo B: apolipoprotein B; HDL-C: high-density lipoprotein cholesterol; LDL-C: low-density lipoprotein cholesterol.

**Table 3 nutrients-15-04036-t003:** Logistic regression analysis of the association between choline and CVD (weighted).

	Model 1		Model 2		Model 3	
Choline	OR (95% CI)	*p*-Value	OR (95% CI)	*p*-Value	OR (95% CI)	*p*-Value
Q1	reference		reference		reference	
Q2	0.904 (0.727, 1.125)	0.359	0.758 (0.583, 0.985)	0.039	0.863 (0.659, 1.131)	0.271
Q3	0.905 (0.739, 1.109)	0.329	0.733 (0.572, 0.940)	0.016	0.848 (0.654, 1.098)	0.200
Q4	0.728 (0.580, 0.914)	0.007	0.611 (0.467, 0.799)	0.001	0.693 (0.520, 0.923)	0.014

Model 1: non-adjusted model; Model 2: adjusted age, gender, race and BMI; Model 3: adjusted age, gender, race, BMI, education level, marital status, smoking status, drinking status, physical activity category, hypertension, DM and PIR. Choline: Q1: < 197 mg/d, Q2: 197–282 mg/d, Q3: 282–392 mg/d, Q4: ≥ 392 mg/d.

**Table 4 nutrients-15-04036-t004:** Association of choline with the total and specific CVD.

Subgroup	OR (95%CI)	*p*-Value
Overall		
Q1	Reference	
Q2	0.863 (0.659, 1.131)	0.271
Q3	0.848 (0.654, 1.098)	0.200
Q4	0.693 (0.520, 0.923)	0.014
CHF		
Q1	Reference	
Q2	0.807 (0.578, 1.127)	0.197
Q3	0.998 (0.668, 1.490)	0.992
Q4	0.743 (0.501, 1.103)	0.133
CHD		
Q1	Reference	
Q2	1.035 (0.733, 1.461)	0.837
Q3	0.874 (0.666, 1.146)	0.314
Q4	0.788 (0.539, 1.151)	0.207
Stroke		
Q1	Reference	
Q2	0.848 (0.609, 1.182)	0.316
Q3	0.858 (0.584, 1.262)	0.421
Q4	0.646 (0.457, 0.913)	0.016

The model was adjusted for the parameters of age, gender, race, BMI, education level, marital status, smoking status, drinking status, physical activity category, hypertension, DM, and PIR. Choline: Q1: < 197 mg/d, Q2: 197–282 mg/d, Q3: 282–392 mg/d, Q4: ≥ 392 mg/d. CHF: congestive heart failure; CHD: coronary heart disease.

**Table 5 nutrients-15-04036-t005:** Subgroup analysis stratified by age, gender, and BMI.

Subgroup	OR (95%CI)	*p*-Value
Overall		
Q1	Reference	
Q2	0.863 (0.659, 1.131)	0.271
Q3	0.848 (0.654, 1.098)	0.200
Q4	0.693 (0.520, 0.923)	0.014
Age < 60 years		
Q1	Reference	
Q2	0.914 (0.609, 1.370)	0.650
Q3	0.706 (0.475, 1.048)	0.081
Q4	0.819 (0.558, 1.202)	0.294
Age ≥ 60 years		
Q1	Reference	
Q2	0.871 (0.640, 1.185)	0.364
Q3	0.931 (0.685, 1.265)	0.634
Q4	0.669 (0.479, 0.934)	0.020
Female		
Q1	Reference	
Q2	0.768 (0.583, 1.011)	0.059
Q3	0.838 (0.605, 1.161)	0.274
Q4	0.788 (0.528, 1.176)	0.232
Male		
Q1	Reference	
Q2	1.079 (0.644, 1.808)	0.764
Q3	0.898 (0.617, 1.307)	0.560
Q4	0.722 (0.453, 1.152)	0.163
BMI < 30 kg/m^2^		
Q1	Reference	
Q2	0.890 (0.642, 1.234)	0.469
Q3	0.794 (0.626, 1.007)	0.056
Q4	0.680 (0.487, 0.949)	0.025
BMI ≥ 30 kg/m^2^		
Q1	Reference	
Q2	0.836 (0.571, 1.226)	0.344
Q3	0.935 (0.587, 1.488)	0.767
Q4	0.742 (0.489, 1.125)	0.152

Logistic association between total choline intake and odds of adjusted age, gender, race, BMI, education level, marital status, smoking status, drinking status, physical activity category, hypertension, DM, and PIR. Choline: Q1: < 197 mg/d, Q2: 197–282 mg/d, Q3: 282–392 mg/d, Q4: ≥ 392 mg/d.

## Data Availability

The data used in this study are publicly available online (https://wwwn.cdc.gov/nchs/nhanes/, accessed on 18 March 2023).

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
