# Peer review of "Association between Dietary Choline Intake and Cardiovascular Diseases: National Health and Nutrition Examination Survey 2011–2016"

_nutrients, 2023, doi:10.3390/nu15184036_

Round 1

Reviewer 1 Report

1. The principal limitation of this study is still the definition at the stage of the diagnosis of CVD. In my opinion, there is a high probability of receiving false information from the interviewee. There are no records in the medical documentation about the presence of CHF, coronary heart disease or stroke. This significantly limits the results of this study.

2. Table 2 shows that the cohort of patients with CVD is statistically significantly older than the Non-CVD cohort of patients, they smoked more often, they had lower eGFR, which indirectly indicates worse kidney function; they were more likely to have hypertension, diabetes, etc. How the authors made adjustment for these CVD risk factors remained unclear to me. I need an explanation.

3. How do the authors explain the results presented in the Table 4? How to explain the significant association of choline levels with stroke, but not with other CVD? After all, CHD, as well as Stroke, has ischemic atherosclerotic pathophysiology?

4. In section 4, the first paragraph.

"Authors should discuss the results and how they can be interpreted from the perspective of previous studies and of the working hypotheses. The findings and their implications should be discussed in the broadest context possible. Future research directions may also be highlighted "

Is this phrase left over from previous reviews? Or is that what the authors really intended?

Author Response

We sincerely thank Reviewer 1 for your very constructive comments. We have made changes according to your suggestions. Our point-by-point responses are listed below:

  1. The principal limitation of this study is still the definition at the stage of the diagnosis of CVD. In my opinion, there is a high probability of receiving false information from the interviewee. There are no records in the medical documentation about the presence of CHF, coronary heart disease or stroke. This significantly limits the results of this study.

Thank you for this very constructive comment! In the NHANES study, trained interviewers used a standardized health questionnaire to survey a nationally representative sample of the US population. The unavoidable recall and report bias may also exist during the process of data acquisition. However, despite this restriction, the general data acquisition procedures in the NHANES database are reliable and convincing, and have been well verified by multiple former research studies.  Previous studies that used NHANES data to explore CVD also used the same definition of CVD (PMID: 35807870, PMID: 36060933, PMID: 36687707).

  1. Table 2 shows that the cohort of patients with CVD is statistically significantly older than the Non-CVD cohort of patients, they smoked more often, they had lower eGFR, which indirectly indicates worse kidney function; they were more likely to have hypertension, diabetes, etc. How the authors made adjustment for these CVD risk factors remained unclear to me. I need an explanation.

Thanks for your detailed review! As we all known, age, smoking status, hypertension, DM and other risk factors have the significant impact on CVD. In the logistic regression analysis, restricted cubic spline regression, and subgroup analysis, we have adjusted risk factors including age, gender, race, BMI, education level, marital status, smoking status, drinking status, physical activity category, hypertension, DM and PIR.

  1. How do the authors explain the results presented in the Table 4? How to explain the significant association of choline levels with stroke, but not with other CVD? After all, CHD, as well as Stroke, has ischemic atherosclerotic pathophysiology?

This is an insightful comment! There are common risk factors and independent risk factors between CVD events and stroke. According to VISP and HPOE2 trails, the reduction of homocysteine could decrease the incidence of stroke events rather than other CVD events (PMID: 18166191, PMID: 17333057). Therefore, homocysteine level is the independent risk factor of stroke. Similarly, we find that dietary choline level is the independent risk factor of stroke, rather than others.

  1. In section 4, the first paragraph.

"Authors should discuss the results and how they can be interpreted from the perspective of previous studies and of the working hypotheses. The findings and their implications should be discussed in the broadest context possible. Future research directions may also be highlighted " Is this phrase left over from previous reviews? Or is that what the authors really intended?

Thank you for the detailed review! This paragraph is a part of Microsoft Word Templates of journal, and it is our fault that we forget to delete it. We have removed this paragraph from our manuscript.

Thanks again for your time and helpful suggestion!

Reviewer 2 Report

The authors analyzed the data from NHNES study for the association between dietary choline and risk for CVD.

Major comments:

Is the “Assessment of dietary choline intake” by 24-hour dietary recall interviews acceptable? Is there a reference?

The study limitation “dietary choline intake information was collected during two 24-hour dietary recall interviews, but individual diet intake may change over time” is the biggest limitation of the study. The manuscript is entirely based on assumed values of dietary choline intake. Neither the dietary intake was known, nor the circulating choline levels were measured. Analysis based on assumption is insufficient to confirm the controversial association between dietary choline and CVD risk. This study merely adds to the controversy without providing evidence.   

The method of “Assessment of dietary choline intake” needs to be included in the abstract.

Minor comments:

Discussion: First paragraph seems to be the reviewer’s comment from the previous submission elsewhere!

Explain p value in Table 1 and mark the significance.

 Minor editing of English language required.

Author Response

We sincerely thank Reviewer 2 for your very constructive comments. We have made changes according to your suggestions. Our point-by-point responses are listed below:

Major comments:

  1. Is the “Assessment of dietary choline intake” by 24-hour dietary recall interviews acceptable? Is there a reference?

We studied many articles about dietary choline intake in the NHANES database, and choose the most reliable assessment of dietary choline. Dietary choline intake from dietary and supplemental sources were combined to generate total intake per day, and an average intake per day was calculated from 1 to 2 recall days depending on data availability (PMID: 34380582).

  1. The study limitation “dietary choline intake information was collected during two 24-hour dietary recall interviews, but individual diet intake may change over time” is the biggest limitation of the study. The manuscript is entirely based on assumed values of dietary choline intake. Neither the dietary intake was known, nor the circulating choline levels were measured. Analysis based on assumption is insufficient to confirm the controversial association between dietary choline and CVD risk. This study merely adds to the controversy without providing evidence.   

Thank you for your valuable comment! There are many studies on dietary factors and cardiovascular diseases based on the NHANES data. The main focus of our study is the association between dietary choline intake and the incidence of CVD risk, rather than other dietary intake, so we just include the dietary choline intake, which is similar with previous dietary intake studies (PMID: 26818246, PMID: 35450998). As you mentioned, while it is important to measure circulating choline levels for our study, NHANES data do not provide this data. Although daily dietary intake may change accordingly with the changes in living conditions, environmental factors and personal habits, it is worth noting that the assessment of dietary intake has been carried out using a 24-hour food recall questionnaire in other comparable studies (PMID: 26818246, PMID: 35450998, PMID: 36769776). This cross-sectional study only suggests that dietary choline intake is associated with the incidence of CVD risk, and no causal inference can be made. Thus, it is also essential to conduct further research studies to monitor and analyze the long-term dietary intake of US individuals.

  1. The method of “Assessment of dietary choline intake” needs to be included in the abstract.

Thank you so much for the constructive comments you raised, and we have added the method of “Assessment of dietary choline intake” in the abstract.

Minor comments:

  1. Discussion: First paragraph seems to be the reviewer’s comment from the previous submission elsewhere!

Thank you for the detailed review. This paragraph is a part of Microsoft Word Templates of journal, and it is our fault that we forget to delete it. We have removed this paragraph from our manuscript.

  1. Explain p value in Table 1 and mark the significance.

Thank you for this very constructive suggestion! The significance of differences between quartiles is indicated by p-value. P-value < 0.05 was considered as statistically significant. As you suggest, we have explained p-value and mark the significant in Table 1.

Thanks again for your time and helpful suggestion!

Reviewer 3 Report

- Authors should better describe and discuss the role of nutraceuticals in managing cardiovascular disease preventive strategies. Please consider and discuss the paper from Scicchitano P et al. Journal of Functional Foods 2014;6:11-32

- use of surveys might be considered as a limitation of the study. Please discuss such a point in a dedicated limitation section.

- authors should discuss about the role of other dietary components in the general assessment of patients and evaluation of the impact of them on final outcomes. Please provide.

- pharmacological background should be described and included in the final regression model. Please provide.

Author Response

We sincerely thank Reviewer 3 for your very constructive comments. We have made changes according to your suggestions. Our point-by-point responses are listed below:

  1. Authors should better describe and discuss the role of nutraceuticals in managing cardiovascular disease preventive strategies. Please consider and discuss the paper from Scicchitano P et al. Journal of Functional Foods 2014;6:11-32

Thank you for this very constructive comment! We have made amendments on our manuscript, and describe and discuss the role of nutraceuticals in managing cardiovascular disease preventive strategies in the manuscript (Line: 224-226).

  1. use of surveys might be considered as a limitation of the study. Please discuss such a point in a dedicated limitation section.

Thank you for this very constructive suggestion! As you suggested, we have discussed this point in the limitation section (Line: 293-296).

  1. authors should discuss about the role of other dietary components in the general assessment of patients and evaluation of the impact of them on final outcomes. Please provide.

Thank you for your valuable comment! There are many studies on dietary factors and cardiovascular diseases based on the NHANES data. The main focus of our study is the association between dietary choline intake and the incidence of CVD risk rather than other dietary intake, so we just include the dietary choline intake, which is similar with previous dietary studies (PMID: 26818246, PMID: 35450998).

  1. pharmacological background should be described and included in the final regression model. Please provide.

Thank you for this very constructive comment! The purpose of our study is to explore the association between dietary choline intake and the incidence of CVD. Pharmaceutical treatment is more closely related to clinical outcomes of CVD. To make our results and conclusions more pertinent to study purpose, we did not involve pharmacological intervention and treatment, and pharmacological background did not be included in the final regression model.

Thanks again for your time and helpful suggestion!

Round 2

Reviewer 1 Report

The authors have made appropriate edits to the article.

Reviewer 2 Report

Nil.